# Rad52 Oligomeric N-Terminal Domain Stabilizes Rad51 Nucleoprotein Filaments and Contributes to Their Protection against Srs2

**DOI:** 10.3390/cells10061467

**Published:** 2021-06-11

**Authors:** Emilie Ma, Laurent Maloisel, Léa Le Falher, Raphaël Guérois, Eric Coïc

**Affiliations:** 1Université de Paris and Université Paris-Saclay, Inserm, LGRM/iRCM/IBFJ-CEA, UMR Stabilité Génétique Cellules Souches et Radiations, F-92265 Fontenay-Aux-Roses, France; emilie.ma@cea.fr (E.M.); laurent.maloisel@cea.fr (L.M.); 2Present address: Precision Oncology Genomics, Oncology Therapeutic Area, Sanofi R&D, F-94403 Vitry-Sur-Seine, France; lea.lefalher@live.fr; 3Université Paris Saclay, CNRS, LBSR/i2BC-CEA, Institute for Integrative Biology of the Cell (I2BC), F-91198 Gif-Sur-Yvette, France; raphael.guerois@cea.fr

**Keywords:** genome stability, DNA repair, Homologous Recombination, Rad52, Rad51, Srs2

## Abstract

Homologous recombination (HR) depends on the formation of a nucleoprotein filament of the recombinase Rad51 to scan the genome and invade the homologous sequence used as a template for DNA repair synthesis. Therefore, HR is highly accurate and crucial for genome stability. Rad51 filament formation is controlled by positive and negative factors. In *Saccharomyces cerevisiae,* the mediator protein Rad52 catalyzes Rad51 filament formation and stabilizes them, mostly by counteracting the disruptive activity of the translocase Srs2. Srs2 activity is essential to avoid the formation of toxic Rad51 filaments, as revealed by Srs2-deficient cells. We previously reported that Rad52 SUMOylation or mutations disrupting the Rad52–Rad51 interaction suppress Rad51 filament toxicity because they disengage Rad52 from Rad51 filaments and reduce their stability. Here, we found that mutations in Rad52 N-terminal domain also suppress the DNA damage sensitivity of Srs2-deficient cells. Structural studies showed that these mutations affect the Rad52 oligomeric ring structure. Overall, *in vivo* and *in vitro* analyzes of these mutants indicate that Rad52 ring structure is important for protecting Rad51 filaments from Srs2, but can increase Rad51 filament stability and toxicity in Srs2-deficient cells. This stabilization function is distinct from Rad52 mediator and annealing activities.

## 1. Introduction

Homologous recombination (HR) is an important pathway of DNA double-strand break (DSB) repair in all kingdoms of life. It is involved in DNA damage tolerance whereby stalled DNA replication forks that have encountered a DNA lesion can resume their progression [1,2]. HR also plays a central role in the correct segregation of homologous chromosomes in the first meiotic division [3], and is implicated in telomerase-independent alternative lengthening of telomeres by which cancer cells avoid telomere degradation [4]. HR uses a sequence homologous to the broken DNA, found preferentially on the sister chromatid or on the homologous chromosome, as a template for DNA repair synthesis [5]. Consequently, HR is a very accurate process. However, its ability to link DNA sequences scattered in the genome might promote genome instability. The two outcomes of repair by HR, gene conversion (GC) and crossing over (CO), are potential sources of important and sudden genetic changes, through rapid transfer of genetic information from one DNA sequence to another and also through genomic rearrangements, such as translocation or repeated sequence shuffling [6]. Additionally, DSB repair mechanisms associated with HR, such as break-induced replication and Single Strand Annealing (SSA), induce the loss of genetic information and can be at the origin of translocations [1,6,7]. Bacterial and yeast models allowed the identification of many control mechanisms to reduce the level of HR-associated genetic instability. For instance, the mismatch repair machinery suppresses heteroduplexes resulting from the interaction between divergent DNA sequences, thus avoiding GC and CO between DNA sequences scattered in the genome [8]. In yeast cells in vegetative growth, HR occurs more frequently through mechanisms that do not give rise to CO. Motor proteins such as Sgs1, Mph1 and Srs2, induce Synthesis-Dependent Strand Annealing (SDSA) by displacing the invading strand from the displacement-loops (D-loops) after DNA synthesis [9,10,11,12,13]. Sgs1, with its partners Top3 and Rmi1, can also dissolve double Holliday junctions as non-CO products [14,15].

The formation of helical nucleoprotein filaments of Rad51/RecA family recombinases on single-stranded DNA (ssDNA) at the lesion site is a key step of HR. These proteins promote the search and strand invasion of a homologous DNA sequence required to initiate DNA repair synthesis [2,16]. In eukaryotes, this process is mediated by the RAD51 recombinase or by DMC1 for meiotic recombination. The formation of RAD51 filaments is tightly regulated to avoid the production of lethal HR intermediates. In yeast cells in vegetative growth, the formation of Rad51 nucleoprotein filaments requires mediator proteins to mobilize the ssDNA binding protein RPA, due to the lower affinity of Rad51 for ssDNA. Rad52 in yeast and BRCA2 in metazoans are among the most studied mediators of Rad51 filament formation [16]. Rad51 nucleoprotein filaments are formed *in vitro* by a two-step mechanism: nucleation of a Rad51 cluster on ssDNA, and cooperative filament growth [17,18,19,20]. Rad51 filament formation also requires Rad51 paralog activity (Rad55/Rad57 in *S. cerevisiae*, RAD51B, RAD51C, RAD51D, XRCC2 and XRCC3 in human cells, and the SHU complex in both), but they might have a more specific role during replication stress [21]. 

In yeast, Rad55/Rad57, the SHU complex, and Rad52 also counterbalance Rad51 filament disruption by Srs2 [22,23,24,25,26,27]. The reason for the regulation of Rad51 filament by positive and negative activities is not well understood, but it is essential to avoid, during HR, the production of lethal intermediates and notably inappropriate Rad51 filaments that are toxic for the cell. The unproductive association of Rad51 with ssDNA might interfere with the normal progression of the DNA replication forks or with DNA repair events [28,29]. Massive Rad52 SUMOylation that leads to the dissociation of Rad52 and Rad51 from ssDNA [30], or Rad52 mutations disrupting the interaction between Rad52 and Rad51 suppress most of the Srs2-deficient cell phenotypes (e.g., DNA damaging agent sensitivity and synthetic lethality upon mutation of genes involved in DNA replication and repair) [26,28]. *Rad55*∆ mutants also can suppress the DNA damaging agent sensitivity in srs2∆ cells (Maloisel, L.; Ma, E.; Coïc E., in preparation). These findings indicate that the Rad52 and Rad55 mediator proteins are directly implicated in Rad51 filament toxicity. Specifically, they stabilize Rad51 filaments to protect them from Srs2 disruptive activity; however, they provide excessive stability to Rad51 filaments in Srs2-deficient cells, thus interfering with other DNA transactions.

As Rad52 is essential at several steps of HR, the stabilization it confers can be observed only by studying separation of function mutations. Here, we describe mutations that are located in the conserved N-terminal domain of Rad52 and that suppress the Srs2-deficient cell phenotype without affecting Rad52 mediator activity. Rad52 N-terminal domain can bind to DNA and carries the catalytic domain for homologous ssDNA pairing [31]. It is also involved in the formation of Rad51 filaments because Rad52 C-terminus, which harbors the RPA and the Rad51 binding domains, is not sufficient for suppressing RPA inhibitory effect in Rad51 filament formation as efficiently as full length Rad52 [32]. To determine how the Rad52 N-terminal domain mutations suppress the Srs2-deficient cell phenotype we performed structural analyses. They suggested that such mutations affect the interaction between the oligomeric ring subunits formed by Rad52 N-terminal domain in solution [33,34,35,36,37]. *In vivo* and *in vitro* analyses of one of these mutations showed that it does not affect Rad52 mediator activity and only slightly its ssDNA binding and homologous ssDNA pairing activity. However, this mutation strongly reduces Rad52 protection of Rad51 filaments against Srs2 destabilization *in vitro*. These observations suggest that the Rad52 N-terminal domain integrity is important for Rad52 stabilization of Rad51 filaments and that this function is distinct from its mediator and annealing activities.

## 2. Materials and Methods

### 2.1. S. Cerevisiae Strains

Strains used in this study are listed in Appendix A. Experiments were mostly carried out in the FF18733 background. Diploid cells used in survival and recombination assays were the result of crosses between isogenic haploid strains bearing the *arg4*-RV and *arg4*-Bg frame-shift mutations. Plasmids bearing the mutations *rad52-V95I*, -*V129A*, -*D79N*, isolated from the random library [26], and *rad52-R37A* were introduced in *rad52*∆ or *rad52*∆ *srs2*∆ cells (MMS experiments). Amino acids are numbered from the first AUG codon in the *RAD52* mRNA [38]. *RAD52-3His-6FLAG* fusion in YCplac111 plasmids [26] with these mutations were also used (γ-irradiation experiments and co-immunoprecipitation experiments). The mutations *rad52-V95I*, *rad52-D79N* and *rad52-R37A* were also introduced in the genome of yeast cells with the pop-in pop-out technique using the integrative plasmids Yiplac211-*rad52-V95I*, Yiplac211-r*ad52-D79N* and Yiplac211*-rad52-R37A* (gene conversion, SSA, ChIP).

### 2.2. Directed Mutagenesis

Single mutations were introduced using a PCR method adapted from [39] in Yiplac211 or Ycplac111 plasmids containing the *RAD52* gene with or without a C-terminal 6His-3FLAG tag.

### 2.3. Sequence Alignment

Homologous sequences of *S. cerevisiae* Rad52 were retrieved using PSI-Blast searches against the nr database [40,41]. The full-length sequences of these homologs were aligned using the MAFFT software [42]. The final alignment was represented using Jalview [43].

### 2.4. Irradiation and Measurement of Recombination Rates

γ-Ray irradiation was performed using a ^137^Cs source. After irradiation, exponentially growing cells were plated at the appropriate dilution on rich medium (YPD) to measure the survival rate, and on synthetic plates without arginine to quantify the number of HR events. The mean percentage of survival from at least three independent experiments is presented.

### 2.5. Survival Following DNA DSB Formation

Cells were grown overnight in liquid culture medium containing lactate before plating. Survival following HO-induced DNA DSB was measured as the number of cells growing on galactose-containing medium divided by the number of colonies growing on YPD. The presented results are the mean of at least three independent experiments.

### 2.6. Structure

The structural model of the Rad52 11-mer was generated using the SWISSMODEL server [44] based on the template of human RAD52 (PDB code: 5xrz) that shares 47% of identity. Conservation was calculated using the multiple sequence alignment of the Rad52 N-terminal domain and the rate4site algorithm [45]. Structures are represented using PyMOL (The PyMOL Molecular Graphics System, Version 2.0 Schrödinger, LLC, New York, NY, USA, 2021).

### 2.7. Cycloheximide Expression Shut-Off Experiment

Cultures grown in YPD medium overnight were diluted to an OD600 value of 0.2 in 30 mL of fresh medium. Cultures were grown at 30 °C to a OD600 value of 0.2 and a 2 mL fraction was removed at the 0 hr time point. Then, cycloheximide (Sigma-Aldrich St. Quentin Fallavier, France) was added to a final concentration of 50 ng/µL. For each time point, OD at 600 nm was measured and a 2 mL fraction was removed. Cell lysis was performed immediately after centrifugation by adding 50 µL of SDS Buffer (50 mM Tris-HCl pH 7.5, 5% SDS, 2.5% glycerol, 50 mM DTT, 5 mM EDTA, 1 x Complete Protease inhibitor cocktail (Sigma-Aldrich, Roche St. Quentin Fallavier, France), bromophenol blue), and boiled at 95 °C for 5 min. Proteins were separated on SDS-PAGE and transferred to a PVDF membrane. Rad52 was detected with a mouse anti-FLAG monoclonal antibody (Sigma-Aldrich, St. Quentin Fallavier, France, 1/10,000) followed by a monoclonal goat anti-mouse IR800 secondary antibody (1/10,000, Advansta San Jose, CA, USA).

### 2.8. Y2H Assay

pBTM116 plasmids carrying *RAD52* (WT or V95I) conjugated with LexA-DBD and pACT2 plasmids carrying *RAD52* (WT or V95I) conjugated with Gal4-AD were transformed in TLM285 cells. pBTM116 plasmids carrying *RAD52* (WT or V95I) conjugated with LexA-DBD and pACT2 plasmids carrying *SRS2* 998–1174 C-terminus conjugated with Gal4-AD were transformed in CTY10-5d cells. Transformants were selected in Synthetic medium without tryptophan and leucine. β-Galactosidase activity was measured with the Yeast β-Galactosidase Assay kit (Thermo Fisher Scientific Rockford, IL, USA).

### 2.9. Co-Immunoprecipitation

Yeast cells were grown in YPD medium to a concentration of 2.5 × 10^6^ cells/mL. Cells were harvested and washed twice with PBS. Extracts were prepared as previously described [46]. Rad52-Rad51 were co-immunoprecipitated as described in [26] and Rad52-RPA and Rad52-Rad59 as described in [28]. Whole-cell extracts (1 mg) were incubated (4 °C for 1 h) with 0.12 µL of anti-Rad51 polyclonal antibody (Abcam Cambridge, UK) for Rad51 immunoprecipitation or with 1µg of anti-FLAG monoclonal (Sigma-Aldrich, St. Quentin Fallavier, France) for Rad52 immunoprecipitation. Then, 50 µL of Dynabeads coupled to Protein A or Dynabeads Pan mouse IgG (Thermo Fisher Scientific, Invitrogen, Waltham, MA, USA) were added, and the incubation continued for another hour. Immunoprecipitates were washed twice with 1 ml of lysis buffer and resuspended in 30 µL of Laemmli buffer (1X). For Rad52-Rad59 co-immunoprecipitation, elution was performed with 50 µg of 3XFLAG peptide to avoid elution of the anti-FLAG antibody, which migrates at the same level as Rad59. The eluted proteins were analyzed by Western blotting. Proteins were separated on 10% SDS PAGE and transferred to PVDF membranes. Proteins were detected with mouse anti-FLAG monoclonal (Sigma-Aldrich, St. Quentin Fallavier, France, 1/10,000), rabbit anti-Rad51 polyclonal (Abcam Cambridge, UK, 1/2000), rabbit anti-RPA polyclonal (a gift from V. Géli, 1/2500), or mouse anti-MYC 9E10 monoclonal (Sigma-Aldrich, St. Quentin Fallavier, France, 1/1000) antibodies. Blots were then incubated with monoclonal goat anti-mouse IR800 or goat anti-rabbit IR700 or IR800 secondary antibodies (1/10,000, Advansta, San Jose, CA, USA). Protein–antibody complexes were visualized using the Odyssey CLx system (Li-cor Biosciences Lincoln, NE, USA). The presence of Rad51 in the immunoprecipitated fractions could not be detected to validate the efficiency of the immunoprecipitation because it migrates at the same level as the anti-Rad51 IgG. However, the absence of Rad52 in the *rad51*∆ immunoprecipitates confirmed that the detected Rad52–FLAG signals were related to the Rad52–Rad51 interaction.

### 2.10. ChIP Experiments and Quantitative PCR Analyses

Cells were grown in YPD until late exponential phase. After inoculation in 400 mL of YPLactate, cultures were grown to a concentration of 5 to 10 × 10^6^ cells/mL. A 50 mL fraction was removed at the 0 h time point, and then galactose was added to a final concentration of 2%. Incubation was continued and 50 mL fractions were removed at different time points. Cells were fixed in 1% formaldehyde, which was then neutralized with 125 mM glycine. Cells were centrifuged and washed with TBS buffer (20 mM Tris pH 8, 150 mM NaCl). Cell pellets were then frozen at −20 °C. ChIP was carried out as previously described [26].

### 2.11. Protein Purification

Rad52-FLAG, Rad52-V95I-FLAG, RPA, Rad51 and His-tagged Srs2 (N-terminal) were purified as in [26].

The Rad52-1-226 fragment, with or without the V95I mutation, was cloned in pCDF-his-SUMO plasmid by SLIC [47]. Rad52-1-226 was purified from *E. coli* BL21 (DE3) cells. Cells were grown in 1 L of LB broth with 50 µg/mL spectinomycin at 37 °C until A600 = 0.8. Protein expression was induced by addition of 0.5 mM IPTG followed by incubation at 37 °C for 4 h. Cells were lysed by sonication in 20 mM Tris-HCL pH 7.5, 500 mM NaCl, 1 mM DTT, 1 mM AEBSF, 10 mM benzamidine, 2 µM pepstatin, 2 µM leupeptin, 1 mg/mL lysozyme and 0.1% NP40. Lysates were clarified by centrifugation and incubated with 20 mM imidazole on Ni Sepharose High Performance Resin (GE Healthcare Velizy-Villacoublay; France) at 4 °C for 4 h. The resin was washed with buffer W1 (20 mM Tris HCl pH 7.5, 500 mM NaCl, 20 mM imidazole, 10% glycerol, 0.5% NP40), and then with buffer W2 (20 mM Tris HCl pH 7.5 at 4 °C, 100 mM NaCl, 20 mM imidazole, 10% glycerol, 1 mM DTT). The His-SUMO tag was cut by addition of SUMO-protease at 4 °C overnight. The Flow-Through was loaded in a 1 mL RESOURCE S column (GE Healthcare Velizy-Villacoublay; France), and eluted through a 25 mL gradient (100 mM-1 M NaCl), in buffer Tris-HCl pH 7.5, 1 mM DTT. The Rad52-1-226-containing fractions were pooled, diluted to a final concentration of 150 mM NaCl, and glycerol was added to 10% final concentration. Rad52-1-226 concentration was determined using an extinction coefficient of 21,430 mole/l/cm at 280 nm.

### 2.12. Electron Microscopy Analysis

For ring analysis, Rad52 was diluted to 2.7 µM in 10 mM Tris pH 7.5, 50 mM KCl, 2 mM MgCl_2_, and 1 mM DTT. Samples were analyzed by conventional electron microscopy using the negative staining method. Three microliters of sample suspension were deposited on an air glow-discharged 400 mesh copper carbon-coated grid for 1 min. The excess liquid was blotted, and the grid rinsed with 2% *w/v* aqueous uranyl acetate. The grids were visualized at 100 kV with a Tecnai 12 Spirit transmission electron microscope (Thermo Fisher, New York, NY, USA) equipped with a K2 Base 4k × 4k camera (Gatan, Pleasanton, CA, USA).

For Rad51 filament transmission electron microscopy studies, a fraction of the following Rad51 filament formation reactions were used. Standard reactions were done by incubating 2.5 µM (nucleotides) viral (+) strand of ΦX174 DNA with 1.2 µM Rad51 (1:2nt) in a buffer containing 10 mM Tris-HCl pH 7.5, 20 mM NaCl, 3 mM MgCl_2_, 1 mM DTT and 1.5 mM ATP at 37 °C for 20 min. Then, 0.17 µM RPA (1:15nt) was added for 10 min. For filament formation in the presence of Rad52, 2.5 µM ΦX174 ssDNA was incubated with 0.17 µM RPA (1:15 nt) at 37 °C in the same buffer for 10 min before addition of 1.2 µM Rad51 (1:2 nt) and 0.17 µM Rad52 (1:15 nt) at 37 °C for 20 min. The size of partial Rad51 filaments ranges from 1500 to 2500 nm, and the size of complete Rad51 filaments is 2800 nm. Srs2 dismantling effect was tested by adding 50 nM Srs2 to the Rad51 filament formation reaction at 37 °C for 5 min. Samples were analyzed by conventional electron microscopy using the negative staining method. Five microliters of sample suspension were deposited on an air glow-discharged 400 mesh copper carbon-coated grid for 1 min. The excess liquid was blotted, and the grid rinsed with 2% *w/v* aqueous uranyl acetate. The grids were visualized at 100 kV with a Tecnai 12 Spirit transmission electron microscope (Thermo Fisher, New York, NY, USA) equipped with a K2 Base 4k × 4k camera (Gatan, Pleasanton, CA, USA).

### 2.13. Electrophoretic Mobility Shift Assay

Increasing amounts of WT and mutant Rad52-FLAG were incubated with 0.27 µM 5′ end-Cy5-labeled XV2 oligonucleotide (5′-TGG GTG AAC CTG CAG GTG GGC AAA GAT GTC CTA GCA ATG TAA TCG TCA AGC TTT ATG CCG TT-3′) in buffer E (10 mM Tris-HCl pH 8, 5 mM MgCl2, 100 mM NaCl) at 37 °C for 10 min. Complexes were separated on 8% native polyacrylamide gels. This experiment was also repeated with dsDNA obtained from annealing 5′ end-Cy5-labeled XV2 with the complementary sequence.

### 2.14. DNA Annealing

Reactions were performed with 200 nM Cy5-labeled Oligo 25 and 200 nM Oligo 26, two 48-nucleotide-long complementary primers described in [48]. Each oligonucleotide was incubated without proteins or with 30 nM RPA at 30 °C for 5 min, before addition of 40 nM of WT or mutated Rad52. An aliquot of the reaction was collected every 2 min and transferred to stop buffer (20 µM unlabeled Oligo 25, 0.5% SDS, 0.5 mg/mL proteinase K). Samples were separated on 8% native TBE polyacrylamide gels. Fluorescent signals were revealed with a Typhoon 9400 scanner and quantified with the ImageQuant TL software.

### 2.15. DNA Strand Exchange Reaction

The 30 µM (nucleotides) viral (+) strand of ΦX174 DNA were coated first with 3.3 µM RPA by incubation in SEB buffer (42 mM MOPS pH 7.2, 3 mM Mg acetate, 1 mM DTT, 20 mM KCl, 25 µg/mL BSA and 2.5 mM ATP) in a final volume of 12.5 µL at 37 °C for 10 min. Rad51 filament formation was initiated by adding 10 µM Rad51 and 15 µM WT or mutated Rad52, or storage buffer as control, at 37 °C for 20 min. Then, addition of 30 µM (nucleotides) of PstI-linearized ΦX174 dsDNA and 4 mM spermidine initiated the strand exchange reaction. After incubation at 37 °C for 90 min, samples were deproteinized by addition of 2 μL of 10 mg/mL proteinase K, 5% SDS solution at 37 °C for 10 min and analyzed by electrophoresis (0.8% agarose gels in 1 × TAE buffer). Standard reactions were done by adding Rad51 prior to RPA. Gels were stained with ethidium bromide and fluorescent signals were imaged with a Typhoon 9400 scanner (Amersham, GE Velizy-Villacoublay; France) and quantified with the Image J software (NIH, USA). The ratio of nicked circular product was calculated as the ratio between the sum of the linear dsDNA substrate and the nicked circular product.

### 2.16. Challenging Rad51 Filaments with Excess Amounts of DNA

Rad52-catalyzed Rad51 filament formation was performed as follows. First, 165 nM RPA (1:15 nt) was incubated with 2.5 μM of 400 nt-long or 1500 nt-long ssDNA (5′ end-Cy5-labeled) in SEB buffer (final volume of 10 μL) at 37 °C for 10 min. After addition of 1.2 μM Rad51 (1:2 nt) and 165 nM Rad52 or Rad52-Y376A (1:15 nt), reactions were incubated at 37 °C for 30 min. Then, ΦX174 viral (+) strand (from 5 to 12.5 μM) was added and incubated at 37 °C for 30 min. Finally, after fixation with 0.25% glutaraldehyde, 4 μL of 40% sucrose was added to facilitate loading on 0.5% agarose gel. After electrophoresis in 1X TAE buffer at 100 V for 1.5 h, the fluorescent signals were imagined with a Typhoon 9400 scanner.

## 3. Results

### 3.1. Mutations Affecting Conserved Residues of the N-Terminal Domain of Rad52 Suppress Rad51 Filament Toxicity in Srs2-Deficient Cells

Looking for *RAD52* mutations that can suppress Srs2-deficient *S. cerevisiae* cells MMS sensitivity, we screened a *RAD52* random mutation plasmid library in cells lacking *RAD52* and *SRS2* (*rad52*∆ *srs2*∆) exposed to methyl methanesulfonate at high dose (MMS, 0.015%). We found three mutations in conserved residues of Rad52 N-terminus: D79N, V95I and V129A (Appendix A). Amino acids are numbered from the first AUG codon in the *RAD52* mRNA [38]. The nucleotide changes are listed in Appendix A. We then inserted these mutations in a centromeric plasmid carrying a *RAD52-FLAG* allele for subsequent *in vivo* analysis by co-immunoprecipitation and chromatin immunoprecipitation (ChIP). Evaluation of resistance to MMS by spot assay of *rad52*∆ *srs2*∆ cells transformed with these plasmids (Figure 1A) showed that each plasmid significantly restored MMS resistance in *rad52*∆ *srs2*∆ cells. We observed this suppressive effect also in γ-irradiated haploid cells where Srs2-deficient sensitivity was totally suppressed by *rad52-V95I*, *rad52-V129A* and *rad52-D79N* (Figure 1B).

We then tested their ability to suppress the synthetic sickness or lethality conferred by Srs2-deficiency in cells in which different genes involved in DNA repair or replication (*rad50*∆, *rad54*∆, *sgs1*∆, *ctf18*∆, *rrm3*∆ or *mrc1*∆) had been deleted. These growth defects are attributed to the formation of toxic Rad51 filaments after the uncontrolled appearance of ssDNA generated by DNA repair or replication defects [28]. Analysis of the meiotic products of diploids resulting from crossing *rad52-V95I srs2*∆ haploid cells with mutant cells showed that *rad52-V95I* suppressed all the synthetic phenotypes (Appendix A).

### 3.2. N-Terminal Rad52 Mutations That Suppress Rad51 Filament Toxicity Are Proficient for HR in Srs2-Deficient Cells

To measure the effect of N-terminal *Rad52* mutations on Rad52 function, we transformed Rad52-deficient cells (but proficient for Srs2) with plasmids harboring the N-terminal *RAD52* mutations. Rad52-V129A and rad52-D79N almost completely restored the resistance to genotoxic agents in *rad52*∆ cells (Figure 1A,B). However, the *rad52-V95I* mutant did not fully rescue MMS sensitivity in *rad52*∆ cells (Figure 1A), and it did not fully complement *rad52*∆ cells for γ-Ray survival, as indicated by the 4-fold survival reduction upon exposure to 400Gy compared with wild type (WT) cells (Figure 1B). Nevertheless, the survival rate of *rad52-V95I* cells was higher than that of Rad52–Rad51 interaction-defective *rad52-Y376A* cells and of Rad52-deficient cells [26]. The absence or relatively low γ-Ray and MMS sensitivity of cells bearing the Rad52 N-terminal mutations suggests that they can form functional Rad51 filaments. This hypothesis was strengthened by the finding that *SRS2* inactivation fully suppressed the γ-Ray and MMS sensitivity of *rad52-V95I* cells (Figure 1A,B). Therefore, as observed for mutations that impair Rad52–Rad51 interaction, the sensitivity of *rad52-V95I* cells to genotoxic agents was not related to reduced Rad52 mediator activity, but was linked to Srs2 activity. Altogether, these results suggest that the Rad52 N-terminal domain stabilizes Rad51 filaments from Srs2 activity, in addition to providing toxicity in Srs2-deficient cells.

Other previously described N-terminal Rad52 mutations show a mild sensitivity to γ-irradiation [49]. This is notably the case for the *rad52-R37A* mutation in Rad52 N-terminal DNA binding domain (initially referred to as *rad52-R70A*) that abolishes the pairing reaction required to complete SDSA [50]. However, this mutation is more sensitive than the mutants we selected and cannot suppress the sensitivity of Srs2-deficient cells (Figure 1B). Therefore, the Rad52 N-terminal annealing activity does not appear to be the main factor of stability provided by Rad52. It is important to note that SRS2 deletion in *rad52-R37A* cells leads to a slight increase in cell viability, suggesting that the R37A mutation might affect Rad51 filament protection against Srs2.

### 3.3. Gene Conversion and SSA Still Occur in the Rad52 N-Terminal Mutants

We then measured the rate of γ-Ray-induced HR between *arg4*-RV and *arg4*-Bg heteroalleles (Figure 1C) in diploid cells that are homozygous for *rad52-V95I* and for *rad52-D79N*, one of the mutations that shows no sensitivity to γ-Ray irradiation (Figure 1D). *Rad52-V95I* homozygous diploids were less sensitive to γ-Ray irradiation than haploid cells. This might be the consequence of the expression of both mating-type alleles *MAT***a** and *MAT*α in diploid cells, which is known to compensate for ionizing radiation sensitivity associated with other mutations, such as *srs2*∆ or *rad55*∆ and *rad57*∆ [51,52,53]. *Rad52-D79N* homozygous diploids were not sensitive to ionizing radiation, and HR frequencies were comparable in mutant and WT cells. However, HR was reduced by 2–3 times in *rad52-V95I* mutants, indicating that although HR still occurred in these cells, Rad51 filament stability or formation was weakened.

Remarkably, the strong hyper-recombination phenotype associated with acute sensitivity to γ-Ray of Srs2-deficient diploid cells was partially suppressed in the *rad52-D79N* and strongly suppressed in the *rad52-V95I* mutant, as observed with the Rad52–Rad51 interaction-defective *rad52-Y376A* mutant [26]. Indeed, *rad52-V95I srs2*∆ homozygous diploid cells were 100-times more resistant than *srs2*∆ cells to 400 Gy, and HR frequency was 100-fold reduced (Figure 1D). This indicates that the Rad52 N-terminal domain is as important as its Rad51-binding domain for Rad51 filament stabilization.

We also measured the effect of both *RAD52* mutants on DSB repair. To this aim, we used a genetic system that allows the repair of a DSB induced by the HO endonuclease at the *MAT* locus of haploid cells (chromosome III) by gene conversion using another *MAT* copy located on chromosome V (Figure 1E) [10]. We observed that the *rad52-V95I* and *rad52-D79N* mutations affected cell survival only marginally, confirming the formation of functional Rad51 filaments in these mutants. As observed before [10], DSB formation strongly reduces survival of Srs2-deficient cells because these cells cannot properly resolve displacement loops through the SDSA pathway. The *RAD52* mutants had only a marginal effect on survival of Srs2-deficient cells, indicating that they cannot rescue the SDSA defect.

Finally, because the N-terminal domain is important for DNA binding and for catalyzing the pairing of homologous ssDNA required to perform SSA, we compared the capacity of *rad52-V95I*, *rad52-D79N* and *rad52-R37A* cells to survive an inducible DSB that can be repaired by SSA between DNA repeats separated by 25 kb [54]. In this system, DSB repair is strictly dependent on *RAD52*. *Rad52-V95I* cells were not sensitive to DSB formation and therefore allowed repair by SSA (Figure 1F). In contrast, the *rad52-R37A* mutant (defective binding to ssDNA) showed a 5-fold reduction in cell survival after DSB formation. Srs2 is also essential to survive DSB formation in this system [54], a result we confirmed here. It has been proposed that Srs2 is required to remove Rad51 that accumulates on long 3′-ssDNA generated from DSB processing [29] and to avoid the formation of branched toxic joint molecules upon ssDNA invasion at ectopic positions [55]. We observed that the *srs2*∆ phenotype was fully suppressed by *rad52-V95I*, partially by *rad52-D79N*, but not by *rad52-R37A*. This result confirmed that Rad52 N-terminus is involved in the toxicity of Rad51 filaments in Srs2-deficient cells, independently of ssDNA binding.

### 3.4. Structural Analysis of the Rad52 N-Terminal Domain Suggests That Rad52-V95I, Rad52-V129A and Rad52-D79N Affect Rad52 Oligomerization

A model of the N-terminal domain of S. cerevisiae Rad52 could be obtained from the oligomeric structure of human Rad52 in its free [36] and bound to ssDNA [56] forms (PDB codes 1H2I and 5XRZ, respectively) (Figure 2A). The crystal structure of the human N-terminal Rad52 revealed a closed ring consisting of eleven monomers. Each monomer has a β-β-β-α fold, made of highly conserved amino acid residues. The subunit contacts involve interactions between β-strands, such that a continuous β-sheet extends around the entire molecule. Additional contacts are provided by the C-terminal helix that swaps across the domain boundary and interacts with an adjacent subunit [33,36]. Structural analysis of this model could be performed for each of the three residues mutated in the screen for *srs2*∆ MMS-sensitivity suppressors (D79, V95 and V129, indicated as pink spheres on the cartoon representation of a monomer of yRad52; Figure 2B) and R37. The conservation map (Figure 2B, lower panel) showed that R37 was the most conserved position, while the other positions exhibited a less stringent conservation pattern in Rad52 homologs. A focus on V95 structural context (Figure 2C, upper left panel) highlighted that its sidechain is fully buried forming a well packed hydrophobic core with a network of hydrophobic sidechains. Mutation V95I could be tolerated assuming that neighboring sidechains adopted different rotameric states to accommodate the extra methyl group, with potential impact on more remote regions, such as those at the interface between Rad52 monomers. Moreover, the bulkier isoleucine residue in the mutant might induce an overpacking of the hydrophobic core that may reduce the intrinsic stability of the folded structure, as observed in other cases of substitutions in the core of folded domains [57]. To test the consequence of a more profound core disruption, we replaced V95 with other residues of different steric and chemical properties. V95T, which conserved the core geometry but reduced its hydrophobicity, behaved as WT Rad52 for MMS resistance and did not suppress the *srs2*∆ mutant MMS sensitivity (Figure 3). Therefore, the slight distortion of the hydrophobic core is important to confer the suppressor phenotype. However, V95D and V95F, which probably disrupt the core, were as sensitive to MMS as *rad52*∆ mutants and did not display any suppressor phenotype (Figure 3). On the other hand, D79 (Figure 2C, upper right panel) lies at the interface between Rad52 subunits and forms an intermolecular salt bridge with K117 in the neighboring subunit. By suppressing the charge complementarity, the D79N mutation may decrease the stability of the salt-bridge interaction, while maintaining the polar contact and bringing minimal steric perturbation. V129 is in contact with I106 in an adjacent Rad52 subunit and might contribute to the stability of the oligomeric Rad52 assembly. The mutation V129A might alter the stability of the inter-subunit Rad52–Rad52 interface by altering the intensity of the hydrophobic effect with neighboring residues. Altogether, a common feature shared by these three positions is that the mutated residues might affect the oligomeric assembly stability rather than directly disturb the interaction with ssDNA. The perturbation caused by each mutation could be of mild amplitude at the level of each individual monomer, but amplified by the oligomeric organization of Rad52, thus cumulating the impact of each monomer. Conversely, the sidechain of R37 directly contacts the phosphate backbone of the ssDNA molecule and this explains the deleterious effect of the R37A mutation on SSA activity.

Our structural studies suggest that Rad52-V95I could affect the oligomer stability. We monitored Rad52 intracellular levels during a cycloheximide expression shut-off experiment and found that the steady-state level of Rad52-V95I, but not of Rad52-D79N, -V129A and -R37A, was lower than that of WT Rad52 (Figure 4A). Rad52-V95I stability *in vitro* was not different compared with WT Rad52 (Figure 4B), suggesting that the destabilization of the hydrophobic core by V95I *in vivo* is the result of increased sensitivity to proteasomal degradation.

As the structural study suggested that all three mutant suppressors affect Rad52 oligomeric organization, we focused on Rad52-V95I that has the strongest impact on Rad52 (Figure 1). First, we wanted to confirm our prediction that the V95I mutation affects Rad52 oligomeric structure. The observation of Rad52 in solution by transmission electronic microscopy (TEM) confirmed that Rad52 formed ring-like structures in solution (Figure 5), as previously reported [35]. With WT Rad52, 87% of the structures observed were well-shaped rings displaying a distinctive hole in the middle. Conversely, with Rad52-V95I, only 72% of rings had a hole, confirming that this mutation slightly destabilizes the ring organization. Surprisingly, we obtained similar results by incubating Rad52 with ssDNA. This effect was more pronounced when we added ssDNA to Rad52-V95I (only 58% of well-shaped rings). Therefore, ssDNA binding and the V95I mutation do not induce the same distortion on Rad52 rings. V95I does not seem to alter the interaction between Rad52 monomers because Rad52-V95I self-interaction was not different compared with WT Rad52, as measured by quantitative two-hybrid screening (Y2H) assay (Appendix A).

### 3.5. Rad52-V95I Affects the N-terminal ssDNA Binding Domain, But Does Not Impair Rad52 Global DNA Binding and Only Marginally Reduces Pairing of RPA-Coated ssDNA

The structural analysis suggested that V95I might not deeply affect Rad52 binding to DNA. Additionally, our genetic analysis showed that it does not alter DSB repair by SSA, suggesting that DNA binding and DNA pairing are not profoundly affected. We used electrophoretic mobility shift assays (EMSA) to test this hypothesis. Purified WT Rad52 and Rad52-V95I (FLAG-tagged) bound similarly to ssDNA and dsDNA (Figure 6A). As Rad52 harbors two DNA binding domains, one in the N-terminal and one in the C-terminal region [32], we asked whether DNA binding through the C-terminal DNA binding domain could hide an N-terminal binding defect conferred by V95I. Therefore, we purified peptides encompassing the WT and V95I-mutated N-terminal domain (1–226), and tested DNA binding by EMSA. The V95I mutation affected both ssDNA and dsDNA binding (Figure 6B), but the mutated N-terminal peptide could still bind to DNA at low concentration (0.05 µM), unlike the 1–326 peptide harboring the R37A mutation that cannot bind to ssDNA at 1 µM [39]. Moreover, full length Rad52-V95I annealed complementary ssDNA strands as efficiently as WT (Figure 6C). Only when the ssDNA strands were coated with RPA, which slows down the reaction catalyzed by Rad52 [47], did we observe a slight decrease (<1.5 times) in the pairing efficiency by Rad52-V95I. In comparison, R37A is totally defective [32].

### 3.6. The V95I Mutation Reduces Rad52 Interaction with Rad59 But Does Not Impair Interaction with RPA, Rad51 and Srs2

The Rad52 N-terminal domain interacts with the Rad52 paralog Rad59 [58], a protein that enhances Rad52 ssDNA annealing [59] and is important for SSA completion [60]. Co-immunoprecipitation experiments showed that compared with FLAG-tagged WT Rad52, the capacity of FLAG-tagged Rad52-V95I to co-immunoprecipitate MYC-tagged Rad59 was reduced by 2-fold (Figure 7A). Therefore, the destabilization of the Rad52 oligomeric structure by the V95I mutation affects Rad59 binding. However, such instability was not strong enough to influence SSA. To rule out that the reduced interaction with Rad59 did not suppress MMS sensitivity in Srs2-deficient cells, we measured MMS resistance in *rad59*∆ *srs2*∆ cells (Appendix A). We observed high MMS sensitivity in *rad59*∆ cells, similar to what observed after γ-irradiation [61]. However, *rad59*∆ *srs2*∆ double mutants were more sensitive to MMS than single *rad59*∆ and *srs2*∆ single mutants, ruling out a *rad59*∆ role as a suppressor.

As the *rad52-V95I* mutation and Rad52 mutations in which interaction with Rad51 is impaired can suppress Srs2-deficient cell phenotypes, we also quantified co-immunoprecipitation of Rad51 with Rad52-V95I-FLAG and WT Rad52-FLAG and we did not detect any difference (Figure 7B). Moreover, the V95I mutation did not affect interaction with RPA (Figure 7C). Finally, we used an Y2H assay adapted from [62] to test the interaction between Rad52 and a peptide covering the 998–1174 C-terminal part of Srs2 (Figure 7D). We found that the V95I mutation did not significantly change binding to Srs2.

### 3.7. The V95I Mutation Does Not Affect Rad51 Filament Formation at a HO-Induced DSB But Increases Rad51 Filament Disruption by Srs2

Our data suggest that altering Rad52 oligomeric structure suppresses the potential toxicity of Rad51 filaments, but also induces the formation of Rad51 filaments that are more sensitive to destabilization by Srs2. This structure modification might have detectable consequences on Rad51 recruitment at DSB. Therefore, we measured the recruitment of proteins involved in Rad51 filament formation by ChIP in haploid cells that express WT or mutant Rad52-FLAG. We used the SSA system designed by Vaze et al. (2002) where a HO-induced DSB can be repaired by SSA between direct repeats located 25 kb apart. This assay involves the formation of long 3′-end ssDNA tails generated from the DSB, thus ensuring the sensitive detection of RPA, Rad52-FLAG and Rad51 recruitment to the DSB site. Quantitative PCR assays using primer sets that amplify DNA sequences at 0.6 kb or 7.6 kb upstream of the DSB site at different time points after DSB induction (Figure 8) showed an increase of the relative enrichment in RPA, Rad52-FLAG and Rad51 at the DSB site compared with the uncut *ARG5,6* locus (blue) in WT cells. At 4 h after DSB induction, RPA, Rad52-FLAG and Rad51 loading was lower at 7.6 kb (green) than at 0.6 kb (red), as described previously [26]. We also confirmed that Rad51 enrichment was higher in srs2∆ than WT cells, particularly at 7.6 kb, showing that Srs2 displaces Rad51 much more efficiently at distant sites [28].

In *rad52-V95I-FLAG* cells, RPA loading was not significantly changed, compared with *RAD52-FLAG* cells. Conversely, Rad51 loading at 0.6 kb was reduced by 2.5-fold. This could be interpreted as a deficiency of Rad52 mediator activity. However, in *rad52-V95I -FLAG*
*srs2*∆ cells, Rad51 loading was 2.8- and 5.5-fold higher than in *rad52-V95I-FLAG* cells at 0.6kb and 7.7 kb respectively, as observed in *srs2*∆ cells. Therefore, the reduction in Rad51 loading in *rad52-V95I-FLAG* cells fully depends on Srs2 activity. We previously obtained similar results with Rad52 mutants that impair Rad52–Rad51 interaction [26]. Rad52-V95I-FLAG recruitment also decreased by 1.7-fold compared with WT, and was significantly increased 1.3-fold in Srs2-deficient cells. Altogether, these results suggested that the Rad52 N-terminal domain integrity is essential to prevent Rad51 and Rad52 disassembly by Srs2.

We also performed the same experiment in *rad52-R37A* cells. Surprisingly, we observed a 3.3-fold decrease in Rad51 recruitment at 0.6 kb, which was Srs2-dependent because Rad51 loading was increased by 2.9-fold in Srs2-deficient cells at this locus. Rad52-R37A-FLAG loading was reduced by 2.6-fold compared with WT. Besides the defect in ssDNA pairing and SSA, Rad52-R37A is also defective for Rad51 filament protection. As Rad52-V95I impairs only Rad51 protection, we concluded that the Rad51 filament protection function can be separated from the ssDNA pairing activity.

### 3.8. The V95I Mutation Abrogates RPA-Mediated Inhibition of Rad51 Filament Formation In Vitro

We then used electron microscopy (EM) to determine the effect of Rad52-V95I on Rad51 filament formation on 5 kb-long ΦX174 viral (+) ssDNA (Figure 9A). As described before (reviewed in [63]), RPA addition to ssDNA prior to the addition of Rad51 inhibits complete Rad51 filament formation, as we previously confirmed [26]. Here, we found that only 24% of complete Rad51 filaments were formed in this condition. The concomitant addition of Rad52 and Rad51 to RPA-coated ssDNA allowed increasing the percentage of complete Rad51 filaments to 78%, confirming that Rad52 overcomes RPA inhibitory effect. Addition of Rad52-V95I, instead of Rad52, to RPA-coated ssDNA also led to the same percentage of complete Rad51 filaments, showing that Rad52-V95I is as efficient as WT Rad52 to overcome RPA inhibition.

These experiments confirmed our previous observation that Rad52 remains associated with Rad51 filaments (Figure 9B) [26,28]. Rad52-V95I also remained associated with Rad51 filaments (Figure 9B). The median distribution of Rad52 and Rad52-V95I associated with Rad51 filaments was the same (1), indicating that the mutation does not modify Rad52 association with Rad51 filaments.

Next, we asked whether Rad52-V95I-catalyzed Rad51 filaments were efficient for strand exchange in an *in vitro* ΦX174-based assay (Figure 9C). As previously described, RPA pre-bound to ssDNA reduced strand exchange by approximately 1.8-fold compared with the standard reaction, where Rad51 is added prior to RPA. The concomitant addition of Rad52 and Rad51 released the inhibition by pre-bound RPA, and led to a 1.6-fold increase in strand exchange. We found that Rad52-V95I could release the RPA inhibition as efficiently as WT Rad52, confirming that destabilization of the Rad52 N-terminal domain does not affect Rad51 filament formation and that these filaments are functional.

Our genetic data and ChIP analysis suggested that V95I affects Rad51 filament protection by Rad52 against Srs2 dismantling activity. We previously reported that this protection could be seen *in vitro* by EM [26]. Therefore, we assembled Rad51 filaments *in vitro* on ΦX174 ssDNA with WT Rad52 or Rad52-V95I, and then incubated them with Srs2 for 5 min (Figure 9D). When Rad52 and Rad51 were added together on RPA-coated ssDNA, about 70% of complete Rad52-associated Rad51 filaments were still present after 5 min of incubation with Srs2. This protection was clearly weakened when Rad51 filaments were formed with Rad52-V95I because we observed only 30% of complete Rad51 filaments after addition of Srs2.

### 3.9. Rad52-V95I Does Not Destabilize Rad51 Filaments in a Competition Assay

Finally, we tested whether Rad52-V95I could challenge the stability of Rad51 filaments by incubation with excess ssDNA (ΦX174 viral (+) strand). We formed Rad51 filaments by adding Rad51 with WT Rad52 or Rad52-V95I on a 400-nt-long or 1500-nt-long Cy5-labeled ssDNA pre-coated with RPA. We used the previously described optimal stoichiometric conditions, with slight modifications [28] (see Material and Methods). Analysis of glutaraldehyde-fixed protein complexes by agarose gel electrophoresis confirmed that Rad52-V95I assembled Rad51 filaments as efficiently as WT Rad52 (Appendix A). After incubation at 37 °C for 20 min to allow Rad51 filament formation, we added the competing ΦX174 viral (+) strand to the reaction for 30 min. The stability of Rad51 filaments assembled with Rad52 or Rad52-V95I was comparable, indicating that Rad51 filament toxicity probably requires additional factors *in vivo*.

## 4. Discussion

### 4.1. Rad52 N-Terminal Domain Integrity Is Required for Rad51 Filament Stability

In a previous study, we described a mutation of Rad52 that could suppress HR toxicity in cells that does not express Srs2, the negative regulator of HR, without affecting the formation of Rad51 filaments [28]. To obtain more insight into Rad52 contribution to Rad51 filament structure and potential toxicity, we then screened a *RAD52* random library for other mutations that suppress HR toxicity of Srs2-deficient cells without affecting the Rad51 filament mediator activity. We found that disrupting the Rad52–Rad51 interaction leads to the suppression of Srs2-deficient cell phenotypes [26]. Here, we describe another set of mutations located in the conserved N-terminal domain of Rad52 that also can suppress the Srs2-deficient cell phenotype.

Like the Rad52–Rad51 interaction mutants, these new mutants can suppress a broad range of *srs2*∆ phenotypes attributed to the formation of toxic Rad51 filaments: MMS and γ-Ray sensitivity in haploid cells, SSA deficiency and synthetic lethality with genes involved in DNA repair or DNA replication [28]. Therefore, losing the Rad52–Rad51 interaction or mutations in the Rad52 N-terminal domain alleviate the toxicity of Rad51 filaments in Srs2-deficient cells. Additionally, the Rad52 N-terminal mutations marginally suppress the *srs2*∆ deficiency in HO-induced ectopic gene conversion, like the Rad52–Rad51 interaction mutants. These phenotypes are probably related to Srs2 post-synaptic deficiency [28]. Therefore, the Rad52 N-terminal domain is directly responsible for the toxicity of pre-synaptic Rad51 filaments in Srs2-deficient cells. Mutations in this domain also displayed defects that are the direct consequence of Srs2 activity. The Srs2-dependent γ-Ray sensitivity and Rad51 binding reduction close to a HO-induced DSB, as observed by ChIP in rad52-V95I cells, show that this mutation makes Rad51 filaments more sensitive to displacement by Srs2. Altogether, these results indicate that the Rad52 N-terminal domain stabilizes Rad51 filaments. Our present and previous results suggest a model in which Rad52 N-terminal domain stabilizes Rad51 filaments by remaining associated with Rad51 filaments through direct interaction with Rad51. Mutations impairing the Rad52–Rad51 interaction and mutations that affect the oligomer stability can bypass the requirement of Srs2 to resist genotoxic agents, while the mutated Rad52 protein still catalyzes the formation of Rad51 filaments ([26] and present results). Rad52 stabilizes and protects Rad51 filaments, making them toxic in the absence of Srs2 activity, probably because they are too stable to be rapidly removed to allow HR progression or the intervention of other DNA repair pathways [64,65]. On the other hand, Rad52 SUMOylation, which triggers the dissociation of Rad52 and Rad51 from DNA [30], can suppress this protection. As Rad52 is essential at several steps of HR, its different effects on Rad51 can be observed only by studying the separation of function mutations. 

### 4.2. Rad52-Dependent Stabilization of Rad51 Filaments Can Be Separated from ssDNA Binding and Homologous ssDNA Pairing Activities

The different phenotypes conferred by the V95I and R37A mutations suggest that stabilization of Rad51 filaments can be separated from homologous ssDNA paring and Rad52 mediator activity.

Our genetic analysis showed that the mutations V129A, D79N and V95I only affect Rad52 capacity to stabilize Rad51 filaments. Our structural analysis suggests that V129 and D79 are located at the interface between each Rad52 monomer and that they are important for the oligomer correct assembly (Figure 2). V95 is fully buried, forming a well packed hydrophobic core with a network of hydrophobic sidechains. Mutation V95I has a potential impact on more remote regions, such as those at the interface between Rad52 monomers. Our EM analysis confirmed that V95I disturbs the correct assembly of Rad52 rings (Figure 5). This mutation also increases Rad52 sensitivity to proteasomal degradation (Figure 4). This observation can explain the strongest phenotype of the V95I mutation observed in γ-irradiated cells compared with V129A and D79N (Figure 1A,B). However, the V129A, D79N and V95I mutations do not affect SSA, indicating that they do not alter Rad52 ssDNA pairing activity (Figure 1F). Moreover, their capacity to repair a DSB shows that they do not affect gene conversion (Figure 1E). Additionally, the V95I mutation does not modify Rad52 capacity to form Rad51 filaments on RPA-covered ssDNA, as observed by ChIP *in vivo* (Figure 8) and by EM *in vitro* (Figure 9). However, the Srs2-dependent γ-Ray sensitivity (Figure 1) and lower recruitment of Rad51 observed by ChIP (Figure 8) indicate that *rad52-V95I* reduces the protection of Rad51 filaments, as confirmed *in vitro* (Figure 9). Surprisingly, ChIP experiments showed that *rad52-R37A* also displays a weaker protection of Rad51 filaments *in vivo*, while it allows the normal recruitment of Rad51 filaments at a DSB (Figure 8). However, unlike the V129A, D79N and V95I mutations, *rad52-R37A* cannot suppress Srs2-dependent γ-Ray sensitivity (Figure 1). This might be related to its default in ssDNA binding and pairing activity [39], making it unable to suppress the Srs2-deficient cell phenotype. Rad59 also could be involved in the protection of Rad51 filaments in conjunction with Rad52, but its strong sensitivity to genotoxic agents might hide this activity, like for *rad52-R37A*. The Rad52 N-terminal mutants we selected display a novel separation of function that allowed us to unravel the role of this domain in Rad51 filament stabilization and protection against Srs2.

### 4.3. How Does Rad52 Stabilize Rad51 Filaments?

Rad52 clearly provides stabilization of Rad51 filaments independently of its mediator activity *in vivo*. Although we did not see any effect of the *rad52-V95I* mutation on Rad51 filament stability in a ssDNA competition assay *in vitro*, as we previously reported for loss of Rad52–Rad51 interaction mutations [26], it is clear that these mutations suppress Rad51 filament toxicity and rescue most of the phenotypes of Srs2-deficient haploid cells. The toxic effect of Rad52 binding to Rad51 filaments *in vivo* might require the association with the yeast Rad51 paralogs Rad55/Rad57 and the SHU complex, because they act as a functional ensemble with Rad52 [66]. Rad52 ring structure could participate in Rad51 filament stabilization through the interaction with Rad51 paralog complexes. It might also provide stability by holding a Rad51 focus on DNA, allowing Rad51 filaments to grow efficiently, as observed for BRC-2 in *Caenorhabditis elegans in vitro* [17].

Rad52 N-terminal domain also inhibits Rad51 filament removal from ssDNA by Srs2. It has been proposed that Rad55/Rad57 and the SHU complex also protect Rad51 filaments from Srs2 activity [22,25]. Rad52 N-terminal domain could counteract Srs2 through a roadblock mechanism, as proposed for the Rad55/Rad57 complex [25]. It could also hold a stable Rad51 nucleus on ssDNA that can provide Rad51 filament elongation despite the Srs2 activity.

## Figures and Tables

**Figure 1 cells-10-01467-f001:**
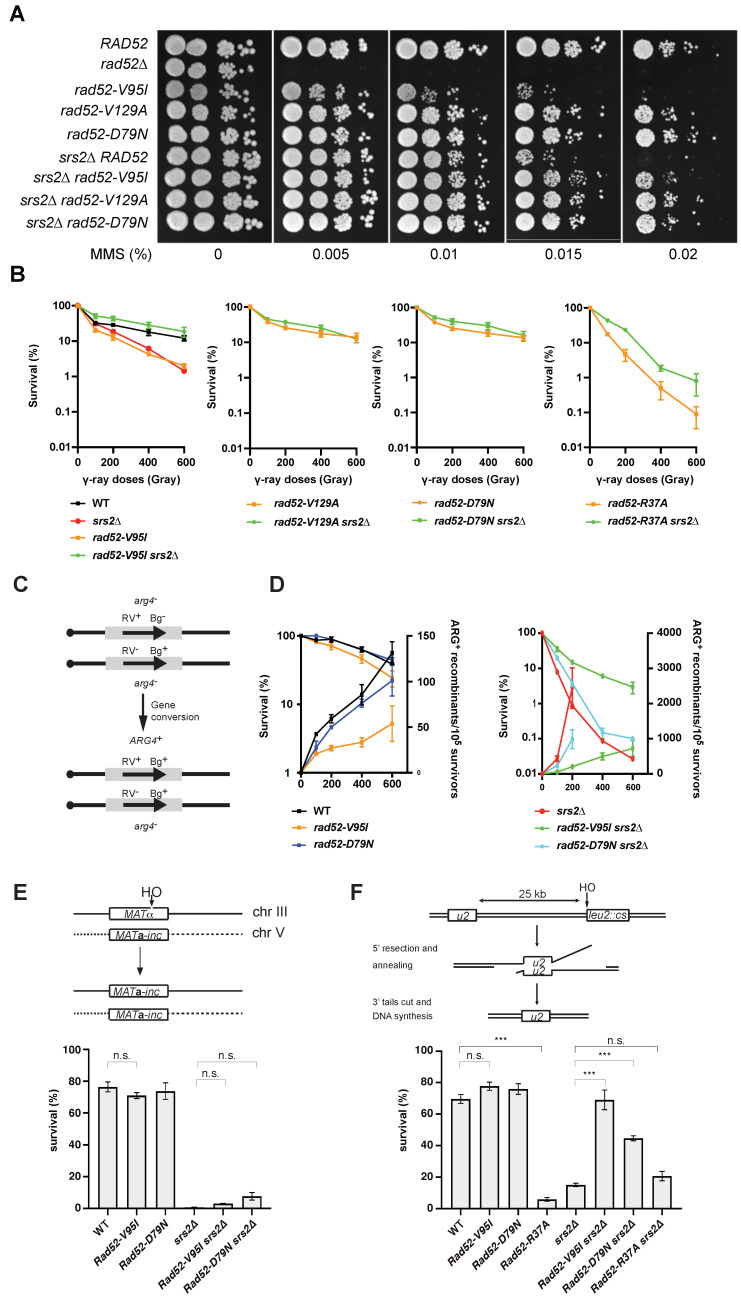
Mutations in the conserved N-terminal domain of Rad52 suppress MMS and γ-Ray sensitivity in Srs2-deficient cells. (**A**) Serial 10-fold dilutions of haploid strains with the indicated genotypes were spotted onto rich medium (YPD) containing different MMS concentrations. (**B**) Survival curves of haploid cells with the indicated genotypes exposed to γ-Ray. (**C**) Recombination system that allows the formation of a WT *ARG4* allele by gene conversion of the *arg4-*RV or *arg4-*Bg heteroalleles in diploid cells. (**D**) Survival curves (left axis) and heteroallelic HR frequencies (right axis) for the indicated homozygous diploid cells exposed to γ-Rays. (**E**) HO-induced gene conversion between *MAT* ectopic alleles. (**F**) Cell survival after HO-induced DSB formation in a SSA repair system. Data are presented as the mean ± SEM of at least three independent experiments; n.s. *p* > 0.05, *** *p* < 0.001 (two-tailed *t*-test).

**Figure 2 cells-10-01467-f002:**
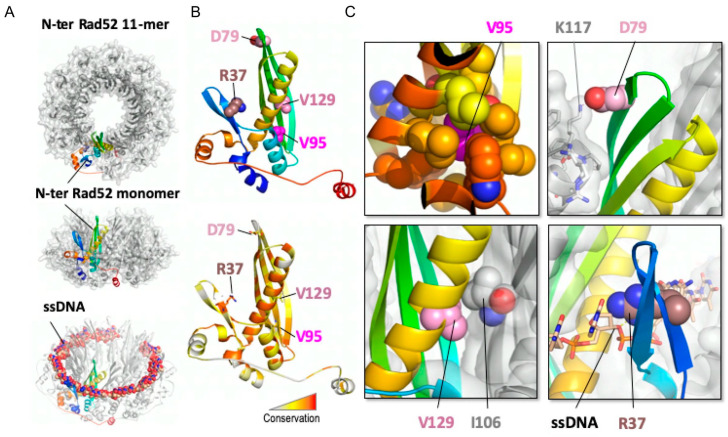
(**A**) Structural representation of the model of S. cerevisiae Rad52 assembled as a 11-mer using the structure of the human homolog bound to ssDNA as template. One subunit is represented as a rainbow cartoon while the other ten subunits are shown as light grey cartoons and transparent molecular surface. (**B**) Cartoon representation of a Rad52 monomer colored in the top panel from N- (blue) to C-terminus (red) and in the lower panel in function of the residue conservation, calculated by the rate4site algorithm with a gradient from white (variable position) to red (conserved position). Residues R37, D79, V95 and V129 are highlighted as pink spheres. (**C**) Four zoomed views centered on each of the four mutated residues highlighting the mutated residue as a sphere. For V95 (upper left panel), the positions surrounding the V95 sidechain (magenta) are shown as spheres with the color code reporting their conservation, as in B (lower panel). For D79 (upper right panel), located at the interface of two subunits, K117 forms a salt-bridge with D79. V129 (lower left panel) is also at the interface between subunits and is highlighted packing against I106 of a neighboring subunit. R37 (lower right panel) contacts the ssDNA phosphate atoms.

**Figure 3 cells-10-01467-f003:**
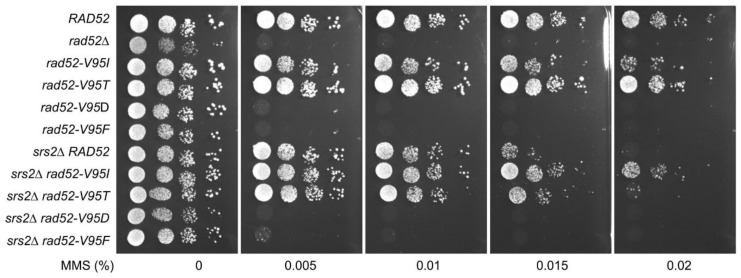
Effect of different V95 mutations on MMS sensitivity and Srs2-deficient cell suppression. Serial 10-fold dilutions of haploid strains with the indicated genotypes were spotted onto rich medium (YPD) containing different MMS concentrations.

**Figure 4 cells-10-01467-f004:**
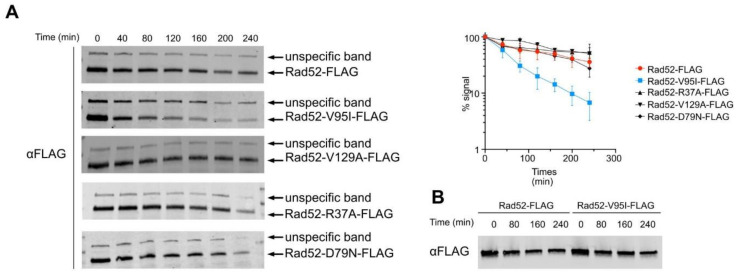
Protein decay of wild-type Rad52 and N-terminal Rad52 mutants. (**A**) Cycloheximide expression shut-off experiments were performed to measure the stability of Rad52-FLAG, Rad52-V95I-FLAG, Rad52V129A-FLAG, Rad52-R37A-FLAG and Rad52-D79N-FLAG. For each time point, the proteins from whole cell extracts were separated by SDS-PAGE and immunoblotted with mouse anti-FLAG monoclonal antibody. The average quantification of at least 3 experiments is shown. (**B**) *In vitro* stability of Rad52-FLAG and Rad52-V95I-FLAG.

**Figure 5 cells-10-01467-f005:**
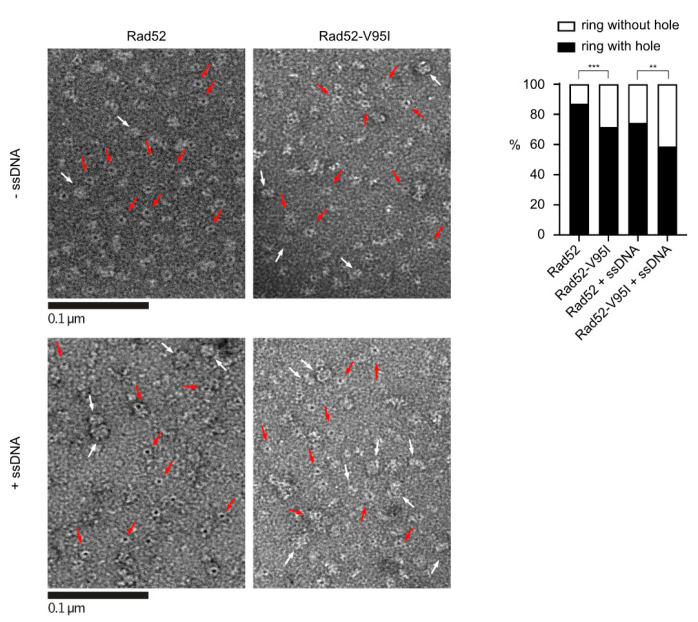
EM analysis of Rad52 and Rad52-V95I without DNA (upper panels) or with 400nt-ssDNA (lower panels). All experiments were performed with FLAG-tagged Rad52 proteins. The protein-ssDNA complex was formed at a ratio of 1:20 nucleotides. Some representative rings with holes (red arrows) and without holes (white arrows) are shown. The percentage of each molecular species is plotted on the histogram (right panel). 641 molecules were analyzed; *** *p* < 0.001; ** *p* < 0.05 (two-tailed Fisher’s exact test).

**Figure 6 cells-10-01467-f006:**
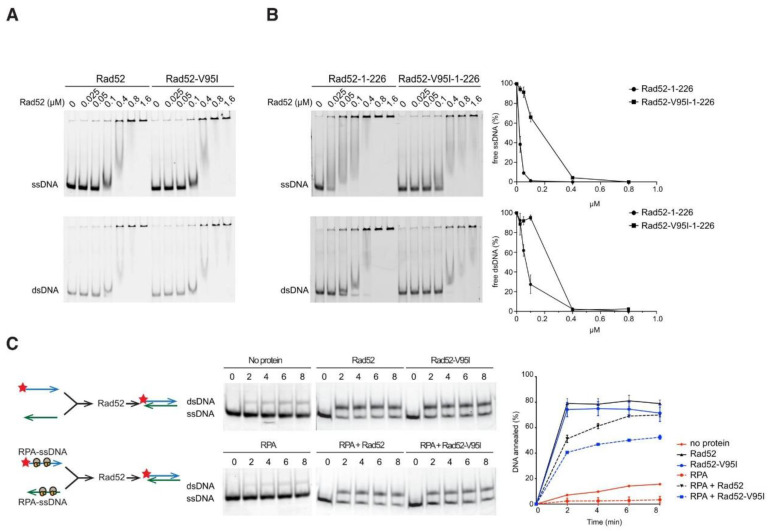
DNA binding and strand annealing activity of Rad52-V95I. (**A**) Binding of purified Rad52-FLAG and Rad52-V95I-FLAG to ssDNA (0.27 µM nt) or dsDNA (0.27 µM nt). (**B**) Same experiment with a peptide covering the N-terminal domain of Rad52 or Rad52-V95I. The quantification of free ssDNA and dsDNA relative to the peptide concentration is shown. (**C**) Rad52-FLAG or Rad52-V95I-FLAG annealing activity of a Cy5-labeled 48 nucleotide-long ssDNA to its complementary primer previously coated or not with RPA. The ratio between annealed DNA and the sum of ssDNA and dsDNA was plotted. Data are presented as the mean ± SEM of at least three independent experiments.

**Figure 7 cells-10-01467-f007:**
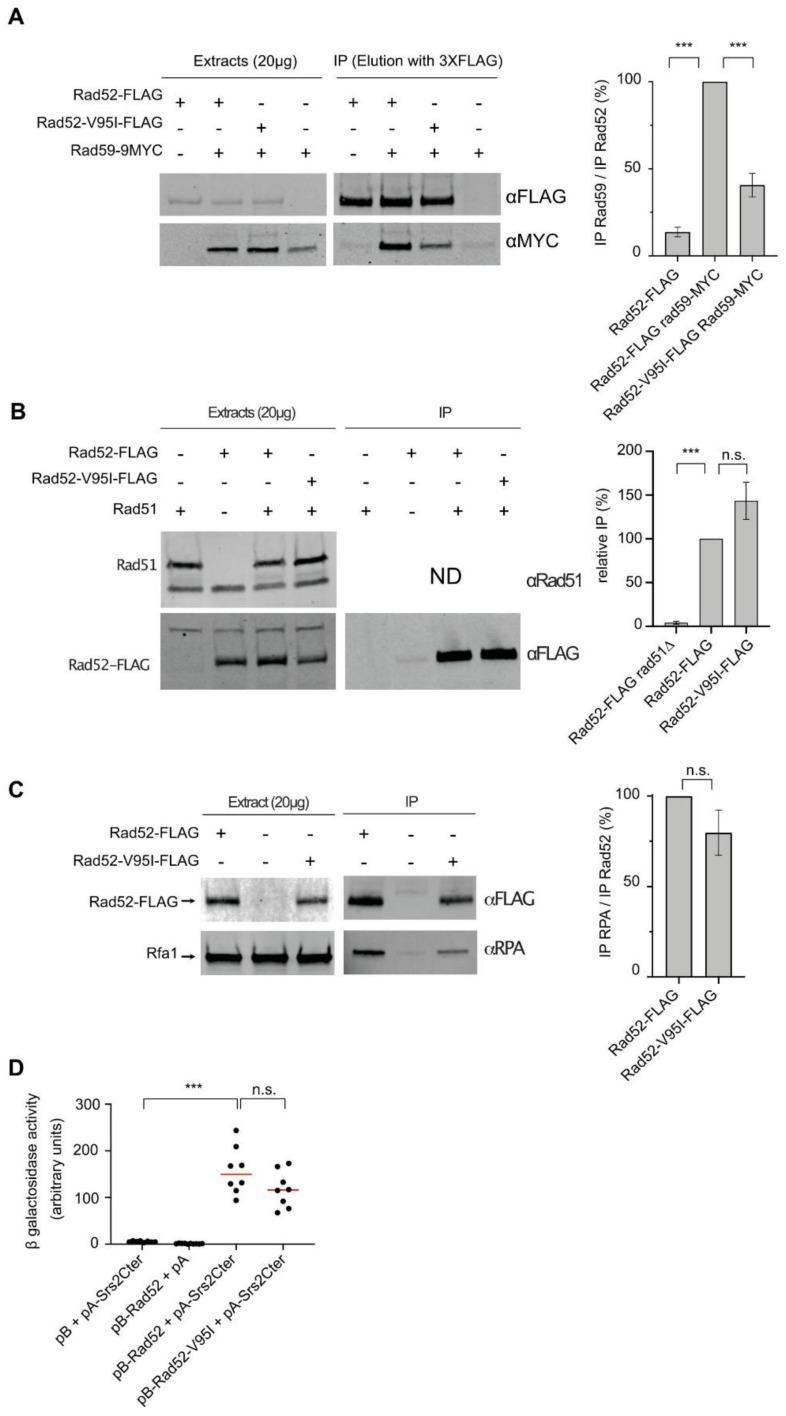
Impact of V95I on Rad52–FLAG interaction with partner proteins. (**A**) Analysis of Rad52-FLAG–Rad59–MYC interaction by co-immunoprecipitation. Rad52-FLAG was precipitated with an anti-FLAG antibody from whole cell extracts. Proteins bound to the antibody were specifically eluted with a 3XFLAG peptide, to avoid the recovery of the anti-FLAG antibody in the immunoprecipitated fraction, because Rad59-MYC migrates at the same level as the anti-FLAG antibody. Rad52-FLAG and Rad59-MYC were detected in whole cell extracts and in the immunoprecipitated fraction (IP) with anti-FLAG and -MYC antibodies, respectively. The histogram shows the ratio between the Rad59 signal relative to the Rad52 signal in the immunoprecipitated fraction (mean ± SEM of at least 2 independent experiments). (**B**) Analysis of Rad51-Rad52-FLAG interaction by co-immunoprecipitation. Rad51 was immunoprecipitated with an anti-Rad51 antibody. The presence of Rad51 in the immunoprecipitated fraction (IP) cannot be detected because it migrates at the same level as the anti-Rad51 IgG used for the immunoprecipitation. However, the absence of Rad52 in the *rad51*∆ immunoprecipitate confirmed that the Rad52-FLAG signals observed are related to the Rad52–Rad51 interaction. The signals corresponding to immunoprecipitated Rad52-FLAG relative to Rad52-FLAG in the whole extract were quantified in two independent experiments and are shown as the mean ± SEM (histogram on the right). (**C**) Analysis of RPA-Rad52-FLAG interaction by co-immunoprecipitation. Rad52-FLAG was precipitated with an anti-FLAG antibody from whole cell extracts. RPA was detected with a polyclonal anti-RPA antibody. The histograms show the ratio between the Rfa1 subunit of RPA signal relative to the Rad52 signal in the immunoprecipitated fraction (mean ± SEM of at least 2 independent experiments). (**D**) Analysis of the interaction between Rad52-FLAG and a Srs2 C-terminal peptide by Y2H analysis. β-Galactosidase activity was measured in CTY10d cells carrying pACT2-GAL4AD (pA)-Srs2Cter and pBTM116-LexABD (pB)-Rad52 or -Rad52-V95I. The median values of β-galactosidase activity are shown (red line). (**A**–**C**) n.s. *p* > 0.05, *** *p* < 0.001 (two-tailed *t*-test), (**D**) n.s. *p* > 0.05, *** *p* < 0.001 (two-tailed Mann and Whitney test).

**Figure 8 cells-10-01467-f008:**
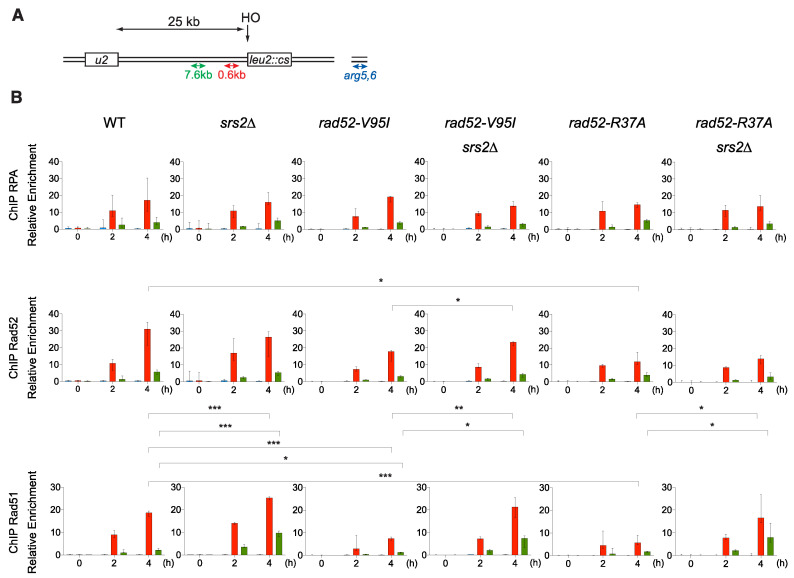
ChIP analysis of Rad51 filament formation at a DSB created by the HO endonuclease. (**A**) Schematic of the HO-induced SSA repair system. The used primers are shown. (**B**) ChIP was used to assess RPA, Rad52 and Rad51 relative enrichment at 0.6 kb (red) and 7.6 kb (green) from the DSB site and at the uncut *ARG5,6* locus (blue) at the indicated time points (hours) after HO induction. The median value (n = 3 or 4) is shown and error bars represent the upper and lower values; * *p* < 0.05, ** *p* < 0.01, *** *p* < 0.001 (unpaired *t*-test for the value at 4 h after HO-induction; only significant differences are shown).

**Figure 9 cells-10-01467-f009:**
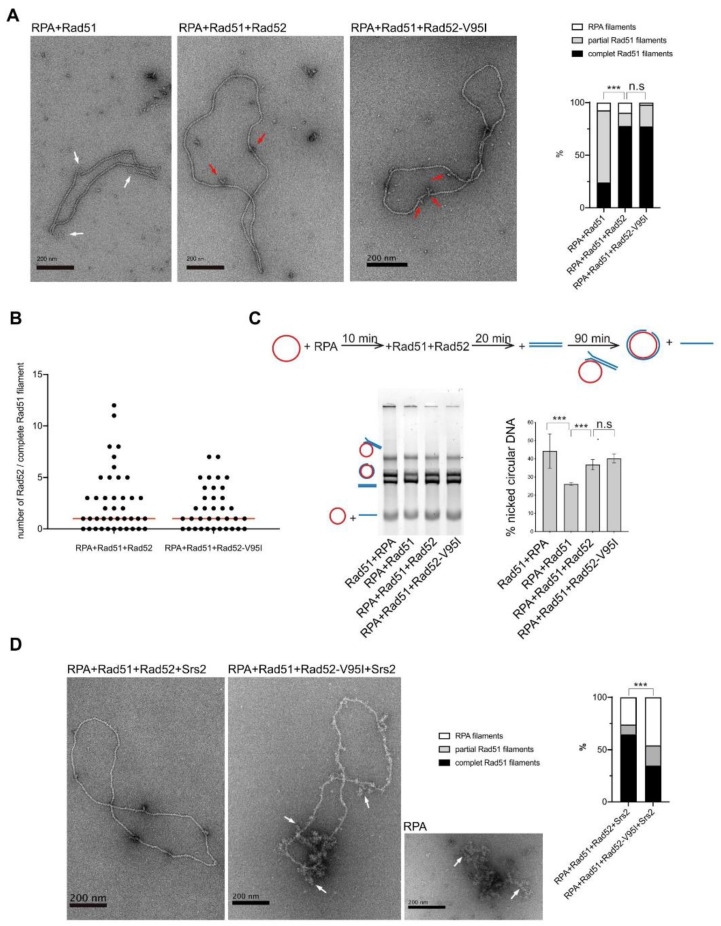
Rad51 filament formation by Rad52-V95I. All experiments were performed with FLAG-tagged Rad52 proteins. (**A**) EM analysis of Rad51 filament formation. A representative nucleofilament issued from each reaction is shown. Addition of RPA before Rad51 (RPA+Rad51) results in the formation of partial Rad51 filaments. White arrows indicate the presence of RPA on ssDNA. Concomitant addition of Rad52 with Rad51 in this reaction (RPA+Rad51+Rad52) leads to complete Rad51 filaments associated with Rad52 (red arrows). The percentage of each molecular species is plotted on the histogram (right panel). Two independent experiments were performed with very similar results. The results of the individual biological replicates were pooled (170 molecules analyzed); n.s. *p* > 0.05, *** *p* < 0.001 (two-tailed Fisher–Freeman–Hamilton exact test). (**B**) Number of Rad52 and Rad52-V95I molecules associated with Rad51 filaments (n > 35). The median value is represented by the horizontal bar; *p* > 0.05 (two-tailed Mann–Whitney test). (**C**) Effect of Rad52-V95I on strand exchange. The standard reaction (Rad51+RPA) shows that complete Rad51 filaments catalyze the formation of nicked circular products. Pre-bound RPA inhibits this reaction (RPA+Rad51); however, the inhibitory effect of pre-bound RPA is overcome by Rad52 and Rad52-V95I (RPA+Rad51+Rad52). The percentage of nicked circular products calculated by quantification of the gels is shown. Data are presented as the mean ± SEM of at least three independent experiments; n.s. *p* > 0.05, *** *p* < 0.001 (unpaired *t*-test). (**D**) EM analysis of Rad51 filament displacement by Srs2. Srs2 was added for 5 min at the end of the Rad51 filament formation reactions. White arrows show the appearance of RPA on Rad52-V95I-assembled Rad51 filaments after Srs2 addition. The percentage of each molecular species is shown in the histogram (right panel). Two independent experiments were performed with very similar results. The results of the individual biological replicates were pooled (160 molecules analyzed); *** *p* < 0.001 (two-tailed Fisher–Freeman–Hamilton exact test).

## Data Availability

Homologous sequences of *S. cerevisiae* Rad52 were retrieved using PSI-Blast searches against the nr database [58,59]. The structural model of the Rad52 11-mer was generated using the SWISSMODEL server [62] based on the template of human RAD52 (PDB code: 5xrz). All data supporting the findings of this study are available within the article and its supplemental data file or from the corresponding author upon reasonable request.

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
