# Peer review of "Rad52 Oligomeric N-Terminal Domain Stabilizes Rad51 Nucleoprotein Filaments and Contributes to Their Protection against Srs2"

_cells, 2021, doi:10.3390/cells10061467_

Round 1
Reviewer 1 Report
Review on Ma et al.
cells-1250041
The paper by Ma et al. describes the isolation and the characterization of a novel mutation in the RAD52 of Saccharomyces cerevisiae, which encodes a protein responsible for various pathways for homologous recombination (HR). The author identified three mutants in the N-terminal domain of Rad51, which plays a role in DNA binding and self-oligomerization, rad52-V95I, -D79N and -V129A, which suppresses a defect conferred by deletion of SRS2, which encodes a DNA helicase dismantling Rad51 filaments. “Stable” Rad51 filaments are toxic so that Srs2 could remove unproductive Rad51 filaments. Previously, the authors showed Rad52 stabilizes Rad51 filament to protect from Srs2. This function is somehow distinct from Rad52’s role in HR as Rad51 mediator. In detailed characterization of Rad52-V95I in both in vivo and in vitro, the authors showed Rad52-oligomer formation is critical for the role of Rad52 as a protection of Rad51 filaments with Rad52 oligomers from anti-recombinase, Srs2. By using molecular genetic, biochemical and electron microscopic analyses, the authors provided large numbers of data sets with a high quality to support the conclusion. The conclusion drawn from the results is very much convincing and warrants for rapid publication.
Although the quantity and quality in the paper are high, Figures needs more works with more cares. For example, there are some many conversion errors in figure captions in PDF such as Fig. 1A, C, E, F, Fig. 3.
Minor points:
- 1D, right graph, caption: Add “I” to rad52-V95 srs2 (rad52-V95I srs2).
- 1B, left graph, Fig. 1D, right graph, caption: Srs2 (red) should be italicized
Author Response
reviewer 1's comment:
The paper by Ma et al. describes the isolation and the characterization of a novel mutation in the RAD52 of Saccharomyces cerevisiae, which encodes a protein responsible for various pathways for homologous recombination (HR). The author identified three mutants in the N-terminal domain of Rad51, which plays a role in DNA binding and self-oligomerization, rad52-V95I, -D79N and -V129A, which suppresses a defect conferred by deletion of SRS2, which encodes a DNA helicase dismantling Rad51 filaments. “Stable” Rad51 filaments are toxic so that Srs2 could remove unproductive Rad51 filaments. Previously, the authors showed Rad52 stabilizes Rad51 filament to protect from Srs2. This function is somehow distinct from Rad52’s role in HR as Rad51 mediator. In detailed characterization of Rad52-V95I in both in vivo and in vitro, the authors showed Rad52-oligomer formation is critical for the role of Rad52 as a protection of Rad51 filaments with Rad52 oligomers from anti-recombinase, Srs2. By using molecular genetic, biochemical and electron microscopic analyses, the authors provided large numbers of data sets with a high quality to support the conclusion. The conclusion drawn from the results is very much convincing and warrants for rapid publication.
Although the quantity and quality in the paper are high, Figures needs more works with more cares. For example, there are some many conversion errors in figure captions in PDF such as Fig. 1A, C, E, F, Fig. 3.
Figures were checked for errors and fixed.
Minor points:
- 1D, right graph, caption: Add “I” to rad52-V95 srs2 (rad52-V95I srs2).
- 1B, left graph, Fig. 1D, right graph, caption: Srs2 (red) should be italicized
1B and 1D was corrected
Reviewer 2 Report
The manuscript entitled ”Rad52 oligomeric N-terminal domain stabilizes Rad51 nucleoprotein filaments and contributes to their protection against Srs2” by Ma and colleagues reports on the regulation of homologous recombination by the balanced actions of Rad52 and Srs2 in stabilizing and destabilizing Rad51 filaments, respectively. The authors have discovered mutations in the monomer-monomer interaction surface of Rad52 that suppresses the DNA damage sensitivity of srs2 mutant cells. This is an interesting finding as it suggests that Rad52 oligomerization is important for stabilizing Rad51 filaments against Srs2 thereby providing another mechanistic level to the regulation of homologous recombination. The manuscript is well written and the data are of high quality and support the conclusions. An few minor corrections and clarifications should be made before publication:
- Figure 1A-B: Why are cells spotted on YPD medium? This would allow the cells to loose the Rad52-expressing plasmid?
- Page 3, line 113: Please provide a reference for the RAD52-3His-6FLAG plasmid or describe its construction.
- Page 9, line 325: change “depleted” to “deleted”.
- Page 10, line 367: it appears from figure 1D that HR was reduced by >10 fold in the rad52-V95I mutant at 600 Gy rather than 2-3 fold as stated in the text?
- Page 15, line 507: change “default” to “defect”.
- Line 588: “ARG5,6” and other genetic elements throughout the manuscript should be italic.
- Line 680: I think this sentence can be misunderstood. Just because toxic recombination by Rad52 can be suppressed by sumoylation of Rad52 independently of Srs2, it doesn’t mean that it is independent of Rad51 filament formation.
- Numbering of Rad52 residues vary in the literature. Please reference the paper that defined the numbering used in this manuscript (Antúnez de Mayolo et al., 2006).
References:
Antúnez de Mayolo, A., Lisby, M., Erdeniz, N., Thybo, T., Mortensen, U.H., and Rothstein, R. (2006). Multiple start codons and phosphorylation result in discrete Rad52 protein species. Nucleic Acids Res. 34, 2587-2597.
Author Response
reviewer 2's comments:
The manuscript entitled ”Rad52 oligomeric N-terminal domain stabilizes Rad51 nucleoprotein filaments and contributes to their protection against Srs2” by Ma and colleagues reports on the regulation of homologous recombination by the balanced actions of Rad52 and Srs2 in stabilizing and destabilizing Rad51 filaments, respectively. The authors have discovered mutations in the monomer-monomer interaction surface of Rad52 that suppresses the DNA damage sensitivity of srs2 mutant cells. This is an interesting finding as it suggests that Rad52 oligomerization is important for stabilizing Rad51 filaments against Srs2 thereby providing another mechanistic level to the regulation of homologous recombination. The manuscript is well written and the data are of high quality and support the conclusions. An few minor corrections and clarifications should be made before publication:
1. Figure 1A-B: Why are cells spotted on YPD medium? This would allow the cells to loose the Rad52-expressing plasmid?
Rad52∆ null mutants display a slight slow growth phenotype on rich YPD medium, a phenotype shared by the Rad52 N-terminal mutant cells. This slow growth phenotype is exacerbated on minimal medium. MMS add to this, making growth of rad52mutant cells and colony formation extremely slow, or even impossible. In order to avoid too much stress to the cells, we test MMS resistance on rich YPD media. The absence of selection for the marker of the plasmid carrying rad52 mutants is largely compensated by the selection pressure applied to these plasmids since they provide resistance to rad52∆ srs2∆ or rad52∆ cells, which are sensitive, or extremely sensitive to MMS respectively.
2. Page 3, line 113: Please provide a reference for the RAD52-3His-6FLAG plasmid or describe its construction.
We provided a reference (now line 114)
3. Page 9, line 325: change “depleted” to “deleted”.
This was changed (now line 329)
4. Page 10, line 367: it appears from figure 1D that HR was reduced by >10 fold in the rad52-V95I mutant at 600 Gy rather than 2-3 fold as stated in the text?
The medium frequency at 600 Gy for WT is 131 ARG+recombinant / 106survivors and 54 for rad52-V95I, for a ratio of 2.4.
5. Page 15, line 507: change “default” to “defect”.
This was changed (now line 515)
6. Line 588: “ARG5,6” and other genetic elements throughout the manuscript should be italic.
This was changed (now line 589). The entire manuscript was scanned for missing italics. We apologize for this.
7. Line 680: I think this sentence can be misunderstood. Just because toxic recombination by Rad52 can be suppressed by sumoylation of Rad52 independently of Srs2, it doesn’t mean that it is independent of Rad51 filament formation.
Massive sumoylation of Rad52 induced by the SUMO-ligase Siz2 overexpression (Esta et al, 2013), or Rad52 mutations that impair Rad52/Rad51 interaction (Ma et al, 2018), or that affect the Rad52 oligomer integrity (this study), all suppressed Srs2-deficient cells MMS sensitivity, and therefore, toxic recombination. This modification and these mutations do not alter Rad51 filament formation as observed both in vivoand in vitro(gene conversion frequencies, ChIP experiments, in vitroassembly of Rad51 filaments). Therefore, we can conclude that Rad52 involvement in Rad51 filament formation can be separated from its contribution to toxicity. We have rearranged the first paragraph of the discussion in a way we think this is now more clear.
8. Numbering of Rad52 residues vary in the literature. Please reference the paper that defined the numbering used in this manuscript (Antúnez de Mayolo et al., 2006).
This was added in the Materials and methods and Results sections (lines 113 and 310)